# Surface-Related Multiples Elimination for Waterborne GPR Data

**Ruiqing Shen [1], Yonghui Zhao [1,\*], Hui Cheng [1], Shufan Hu [2], Shifeng Chen [3] and Shuangcheng Ge [4]**

1   School of Ocean & Earth Science, Tongji University, Shanghai 200092, China; stricky_srq@tongji.edu.cn (R.S.); 2111322@tongji.edu.cn (H.C.)
2   School of Mathematics and Computer Sciences, Nanchang University, Nanchang 330047, China; shufanhu@ncu.edu.cn
3   Zhejiang Huazhan Engineering Research and Design Institute Co., Ltd., Ningbo 315000, China; shifchen1974@163.com
4   School of Water Conservancy and Environment Engineering, Zhejiang University of Water Resources and Electric Power, Hangzhou 310018, China; gesc163@163.com
\*   Correspondence: zhaoyh@tongji.edu.cn

**Abstract:** Ground-penetrating radar (GPR) is a well-respected, effective, and efficient geophysical technique. However, for underwater engineering detection and underwater archaeology, the measured B-scan profiles typically contain surface-related multiple waves, which can reduce the signal to noise ratio and interfere with the interpretation of results. SRME is a feedback iteration method based on wave equation, which is frequently utilized in marine seismic explorations but very rarely in GPR underwater engineering detection. To fill this gap, we applied SRME to suppress multiples that appear in GPR underwater images. When we compared the effectiveness of the underwater horizontal layered model and the underwater undulating interface model, we found a high match rate between the predicted and the real-world multiples. In addition, the addition of the Gaussian random noise level with a 4% maximum amplitude to the B-scan profile of the horizontal stratified model yielded satisfactory multiple suppression results. Finally, we applied this method to the B-scan GPR section of actual underwater archaeological images to achieve multiple suppression, which can more effectively weaken and inhibit the surface-related multiples. Both numerical simulations and actual field data show that the SRME method is highly suitable for interpreting waterborne GPR data, and more accurate interpretation can be obtained from the GPR profile after multiples suppression.

**Keywords:** waterborne GPR; multiples; surface-related multiples elimination; underwater detection

## 1. Introduction

Ground-penetrating radar (GPR), a near-surface geophysical method, is frequently utilized in many fields including geological prospecting [1], engineering detection [2,3], and archaeological prospecting [4,5]. Underwater engineering exploration focuses on underwater topography, silt layer, and riprap [6–9]. GPR has been evaluated as a tool for mapping lake bottoms and ice-thickness [10–15]. For example, Moorman and Michel [16] were able to measure water depths and lacustrine sediment thickness of Artic lakes with precisions of ±3% and ±15, respectively, via bathymetric mapping with this method. They found that lake bottom multiples could only affect interpretations with good sub-bottom penetration or shallow water depth. Dugan et al. [17] used GPR and airborne transient electromagnetic (AEM) surveying to map the ice stratigraphy and ice-bed contact of Lake Vida and locate and delineate a confined aquifer with a porosity of 23–42%. Bandini et al. [18] found that water coupled and a drone borne GPR antenna were both effective for bathymetry of inland bodies of water. Due to its high resolution and adaptability, GPR has also been successfully applied in underwater archaeology by providing researchers with accurate spatial distribution information of underwater cultural relics [19]. These application studies and

experiments on real world data using various water detection methods indicate that GPR can effectively reveal underwater targets in freshwater environments. However, according to the results of real-world exploration, multiple waves in the underwater profiles detected by GPR can reduce the signal-to-noise ratio and severely interfere with the identification of effective waves, and even lead to false imaging [20,21].

Currently, studies of multiples suppression of GPR data mainly emphasize detection on land. Zhao et al. [22] tested predictive deconvolution, F-K transformation, and the ability of Karhunen–Loeve transformation to suppress multiple diffraction waves of steel bars in shield tunnel lining segments to observe the effect of grouting behind the wall, which yielded unsatisfactory results. Xie et al. [23] utilized the predictive deconvolution method to suppress the multiple waves in the GPR profiles of reinforced concrete structures, and the result indicates that the multiple waves from steel bars have been obviously removed. Zhang and Slob [24] proposed a one-dimensional method based on the electromagnetic Marchenko equation for eliminating multiples hierarchically, which would allow conventional imaging methods to obtain imaging results without interval multiples. The method works well for a sample in a synthetic waveguide.

Presently, the surface-related multiples elimination (SRME) method has rarely been applied to underwater GPR data. To fill this gap, only a small number of researchers have used this method in marine seismic explorations. Berkhout [25] proposed a feedback technique for complex multiple wave systems that uses a multidimensional iterative inversion algorithm to eliminate free surface multiple waves. This model can be adapted to any complex underground medium and considers the characteristics of the source and detector. Thus, it lays the theoretical foundation for feedback iterative multiple suppression methods. Inspired by the pre-stacked inversion theory, Verschuur et al. [26] successfully eliminated the predicted free surface multiples according to the principle of minimum energy and was the first to utilize the adaptive subtractive surface operator calculation method, allowing SRME to be applied to the processing of real-world data. As the estimation of surface operators is a nonlinear problem, Berkhout and Verschuur [27] further proposed an iterative algorithm for predicting multiples, of which the iterative process converges quickly, and the errors generated by one iteration will affect data in the next iteration. For further elimination of surface-related multiples, Verschuur and Berkhout [28] found that the SRME method can also perform local elimination within the local window after global elimination has taken place. Jakubowicz [29] proposed a data-driven interval multiples suppression (IMP) technique based on the SRME method that does not require reconstruction of the model reference plane. Berkhout and Verschuur [30] extended the free surface multiples to the interval multiples and proposed the generalized SRME method depending on the velocity model.

In this paper, we first used the SRME method to process waterborne GPR data. Two numerical model experiments were carried out to verify its ability to suppress surface-related multiples. In order to determine the effectiveness of multiple suppression by SRME, we compared it to the prediction deconvolution method.

## 2. Materials and Methods

### 2.1. Principles of the SRME Method

According to Verschuur [31], a source signal $s(t)$ propagates downwards vertically. The primary wave $y_0(t)$ is calculated by the equation below:

$$y_0(t) = s(t) * x_0(t) \tag{1}$$

where $x_0(t)$ represents the formation responses. When the primary wave $y_0(t)$ is reflected at the surface, the first-order surface-related multiple can be expressed as:

$$m_1(t) = -y_0(t) * x_0(t) \tag{2}$$

for which the minus denotes the total reflection. Similarly, the *n*-order surface-related multiple can be written as follow:

$$m_n(t) = -m_{n-1}(t) * x_0(t) \tag{3}$$

Therefore, the total responses can be calculated by the equation below:

$$y(t) = y_0(t) - y_0(t) * x_0(t) + y_0(t) * x_0(t) * x_0(t) - \cdots \tag{4}$$

Equation (4) means that all responses are the result of repeated multiples. This equation can be also expressed as follows:

$$y(t) = [s(t) - y(t)] * x_0(t) = y_0(t) - y(t) * x_0(t) = s(t) * x(t) \tag{5}$$

where $x(t)$ is the total of all the formation responses. In practice, $x(t)$ is unknown. To estimate the formation responses, a deconvolution operator is defined as

$$a(t) * s(t) = \delta(t) \tag{6}$$

where $a(t)$ is the deconvolution operator and $\delta(t)$ is a pulse signal. According to Equations (1), (5) and (6), the frequency domain of the primary wave can be expressed by the following equation:

$$Y_0(f) = \frac{Y(f)}{(1 - X(f))} \tag{7}$$

Equation (7) can be transformed through polynomial expansion as follows:

$$Y_0(f) = Y(f) + A(f)Y^2(f) + A^2(f)Y^3(f) + \cdots \tag{8}$$

According to Berkhout and Verschuur [27], Equation (8) can be changed into the equation below:

$$Y_0{}^{i+1}(f) = Y(f) - A(f)Y_0{}^i(f)Y(f) \tag{9}$$

where the superscript *i* denotes the iterations. Usually, the first iteration input is the total responses $Y(f)$. The deconvolution operator $A(f)$ or $a(t)$ must be calculated at every iteration according to the principle of minimum energy. In the time domain, Equation (9) is expressed as follows:

$$y_0^{i+1}(t) = y(t) - a(t) * y_0^i(t) * y(t) = y(t) - a(t) * m^i(t) \tag{10}$$

in which $m^i(t)$ represents the predicted multiples throughout the *i* iterations. According to the principle of minimum energy, the energy can be calculated using the equation below:

$$E = \sum_{n=0}^{M} \left\{ y(n) - \sum_{k=0}^{N} a(k)m(n-k) \right\}^2 \tag{11}$$

where *M* and *N* are the length of the signal and the length of the deconvolution operator $a(t)$, respectively. Energy (*E*) can be regarded as the function of $a(k)$. The differential derivative of E with respect to each $a(k)$ is zero so that E reaches the minimum. Finally, the equation of $a(t)$ can be written as

$$\sum_{k=0}^{N} \phi_{mm}(j-k)a(k) + \varepsilon^2 a(j) = \phi_{ym}(j); j = 0, 1, 2, \ldots, N \tag{12}$$

where $\phi_{mm}$ denotes the autocorrelation of the predicted multiples, $\phi_{ym}$ represents the cross-correlation of the total of input responses and the predicted multiples, $\varepsilon^2$ is the stability

coefficient. The left side of Equation (12) can be considered a Toeplitz matrix. Levinson [32] proposed an efficient recursive algorithm for solving such a matrix.

### 2.2. Workflow

The flowchart of SRME is shown in Figure 1. The first step consisted of pre-processing procedures including correcting for time-zero, muting of direct waves, establishing an automatic gain control (AGC), filtering, and denoising. Filtering methods were used to suppress ambient noise, which usually includes removing the global background and filtering the finite impulse response (FIR) filter. In this paper, we utilized the denoising method to eliminate the influence of residual noise in bodies of water. These data were substituted by weak random noise. The second step was to obtain the surface-related multiples prediction model through autocorrelation of the input signal, i.e., the surface-related multiples prediction model. The third step was to obtain self-adaptive surface matching operators based on the principle of minimum energy within the global time window. The critical parameters include the global operator length, the iteration time, and the white noise coefficient, i.e., the stability coefficient. For our study, the best global operator length was not fixed and was no more than half the time window. There were generally no more than five iterations. The value of the white noise coefficient varies from 0.1% to 1%, and had very little impact on the SRME results. Finally, we utilized the iterative subtraction method to obtain the surface-related multiples elimination results.

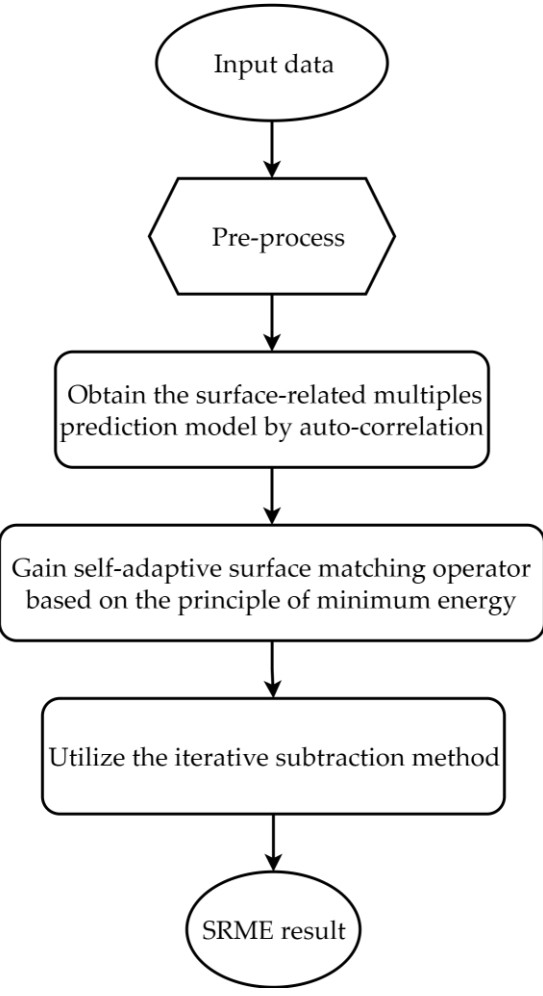

**Figure 1.** The flowchart of the SRME method.

### 2.3. Synthetic Datasets

2.3.1. Underwater Horizontal Layered Model

Two groups of synthetic datasets testing were carried out to validate the effectiveness of the SRME method within an FDTD algorithm [33]. Figure 2a shows a simple underwater horizontal four layer model, the parameters of which are shown in Table 1. The excitation and reception antennas were placed at the top in the air layer, 0.01 m away from the surface of the water. We also used the Ricker wavelet and the 100 MHz center frequency of the electromagnetic wave. Each grid of the model was 2 mm, and the PML absorption boundary was set to 40 grid widths or 8 cm. The excitation and receiving antennas moved 2 cm to the right at a horizontal step of 2 cm from 0.1 m and 0.14 m, respectively. A total of 88 channels of data were collected, within the 85 ns recording window. Figure 2b shows the raw B-scan profile. Figure 2c illustrates the processed version in which the direct waves were muted. Figure 2d shows the B-scan profile with a Gaussian random noise level of 4% of the maximum amplitude based on Figure 2c.

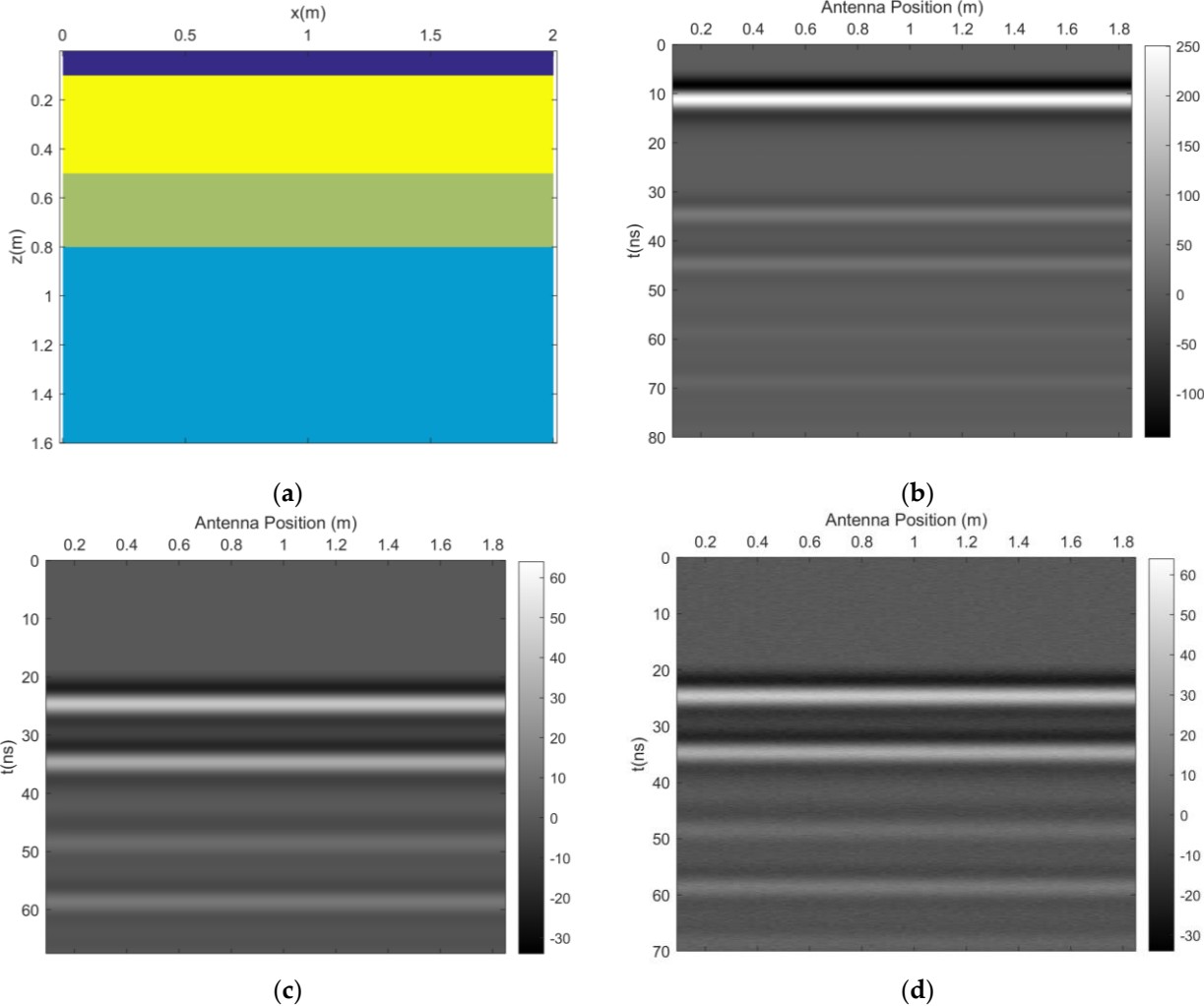

**Figure 2.** The horizontal layered model and forward results: (**a**) The underwater horizontal layered model; (**b**) The raw B−scan profile; (**c**) The B−scan profile in which direct waves have been muted; (**d**) The B−scan profile with a Gaussian random noise level of 4% of the maximum amplitude based on (**c**).

**Table 1.** Material parameters of the horizontal layered model.

| Layer | Thickness (m) | Relative Permittivity $\varepsilon_r$ |
|---|---|---|
| Air | 0.1 | 1 |
| Water | 0.4 | 81 |
| Silt | 0.3 | 25 |
| Bedrock | 0.8 | 5 |

2.3.2. Underwater Undulating Interface Model

Figure 3a shows a 9 m × 3 m underwater undulating interface model with a grid width of 5 mm. The excitation antenna was set at the horizontal location of 0.3 m, the receiving antenna at 0.4 m, and both were 0.05 m away from the surface of the water. The horizontal step was 0.1 m. A total of 84 channels of data were recorded during a window of 145 ns with a center frequency of 100 MHz. The model parameters are given in Table 2, in which the material characteristics correspond with those in Table 1. The maximum and minimum depths of the water layers were 1.1 m and 0.7 m, respectively. The maximum and minimum thicknesses of the silt layer were 1.5 m and 0.5 m, respectively. Figure 3b,c show the raw B-scan profile and the processed version. Figure 3d shows the B-scan profile with a Gaussian random noise level of 4% of the maximum amplitude based on Figure 3c. According to the identification characteristics and periodicity of multiples, the corresponding event, indicated by the yellow arrow, is a result of the second-order surface-related multiples of the primary reflection at the bottom. The dip angle of the multiple in the curved section increased compared to that of the primary wave. The blue arrow in the middle represents the short-range surface-related multiples in the silt at the bottom, of which the travel time is identical to that of the two yellow arrows in the middle. A relatively insubstantial event occurred near the concave center at 100 ns. According to the travel time, there were multiple intervals between the water-subsurface and the silt-subsurface.

**Table 2.** Material parameters of the underwater undulating model.

| Layer | Thickness (m) | Relative Permittivity $\varepsilon_r$ |
|---|---|---|
| Air | 0.3 | 1 |
| Water | 0.7 (thinnest)/1.1 (thickest) | 81 |
| Silt | 0.5 (thinnest)/1.5 (thickest) | 25 |
| Bedrock | 1.1 | 5 |

*2.4. Real Case Datasets*

The real case datasets came from an underwater archeological site at Shanglin Lake in Cixi City, Zhejiang Province, featuring ancient Yue kilns that were in use for over 1000 years from the Eastern Han Dynasty to the middle of the Southern Song Dynasty. Nearly 200 sites have been discovered, 179 of which have been numbered. It is currently the largest group of celadon firing sites and the most concentrated distribution of kilns in China. Figure 4 shows Shanglin Lake and its surroundings, particularly the water-covered Houshi'ao site, indicated by the red circle. This site is located in the southern region of the west bank of the Shanglin Lake Reservoir, which has one of the highest distributions of kilns. However, due to the limited excavation area and lack of underwater archaeological investigations, the exact range, shape, and accumulation of these underground objects cannot be accurately defined, and the submerged remains have not yet been explored.

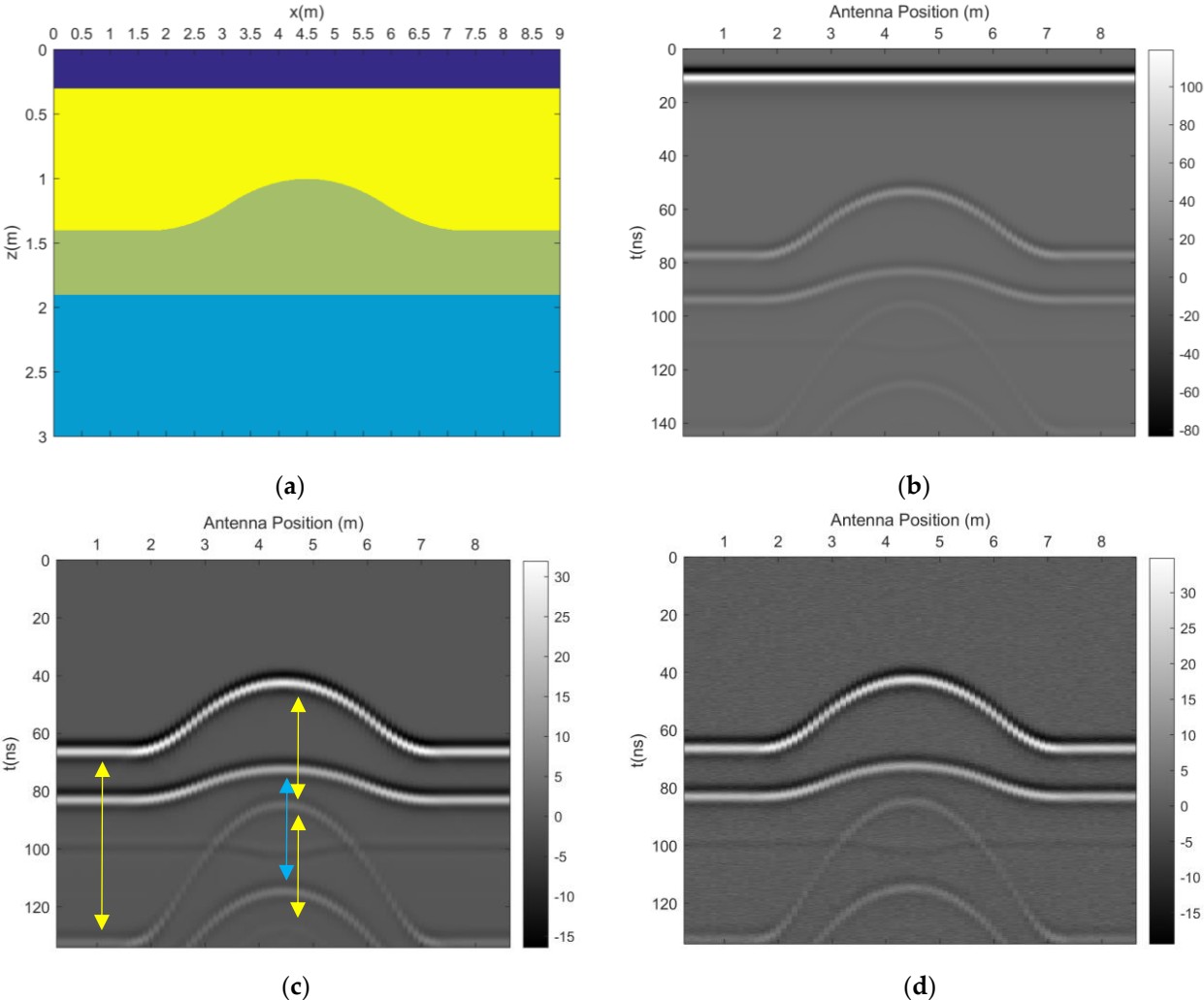

**Figure 3.** The underwater undulating model and forward results: (**a**) The underwater undulating model; (**b**) The raw B−scan profile; (**c**) The processed B−scan profile, in which the yellow arrow indicates the second−order surface−related multiples of the primary reflection at the bottom, and the blue arrow represents the short−range surface−related multiples in the silt at the bottom; (**d**) The B−scan profile with a Gaussian random noise level of 4% of the maximum amplitude based on (**c**).

Based on prior detection results, Shanglin Lake can reach a maximum depth of 8 m. The GSSI SIR-4000 GPR with a center frequency of 100 MHz was chosen for Houshi'ao site, as shown in Figure 5. The average relative dielectric constant and conductivity of the water in this area were 83 and 0.003 S/m, respectively. Therefore, the propagation speed of electromagnetic waves was estimated to be about 0.033 m/ns. Similarly, the estimated velocity of electromagnetic wave propagation in the bottom layer medium of the lake was 0.055 m/ns. The length of this survey line was 184 m. The measurement carrier is a diesel-powered wooden boat, of which the thickness of the bottom plate is 5 cm. We used the common offset acquisition method based on the time-sequence with a scanning rate of 24 scan/s, a sampling rate of 2048, and a time window of 800 ns. The raw datasets contain 4459 channels of A-scan data. The raw and pre-processed B-scan profiles are shown in Figure 6a,b, respectively. The pre-processing procedures included Dewow and FIR filtering, AGC, time zero correction, the removal of DC components and direct waves as well as water noise suppression. Compared to Figure 6a, the data that was collected above the underwater reflection interface in Figure 6b was filled with weak random noise, which can reduce the impact of water noise. The B-scan profile contained well-developed multiples and four distinct rises. The flat bottom is mainly located at about

200 ns, and the multiples at 400 ns are the first order surface-related multiples. Due to the attenuation of electromagnetic waves during propagation, the events of the flat formations at around 300 ns are blurry.

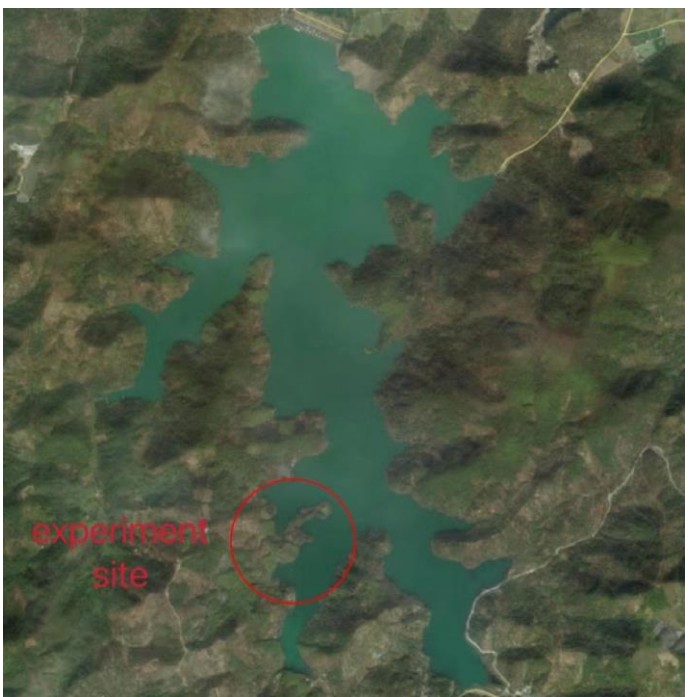

**Figure 4.** The location of the underwater archeological site atShanglin Lake.

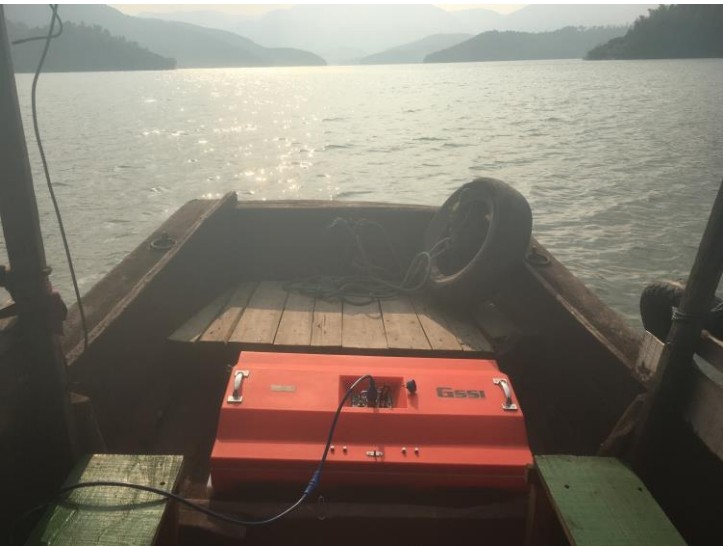

**Figure 5.** An on-site photo using GSSI SIR-4000 GPR.

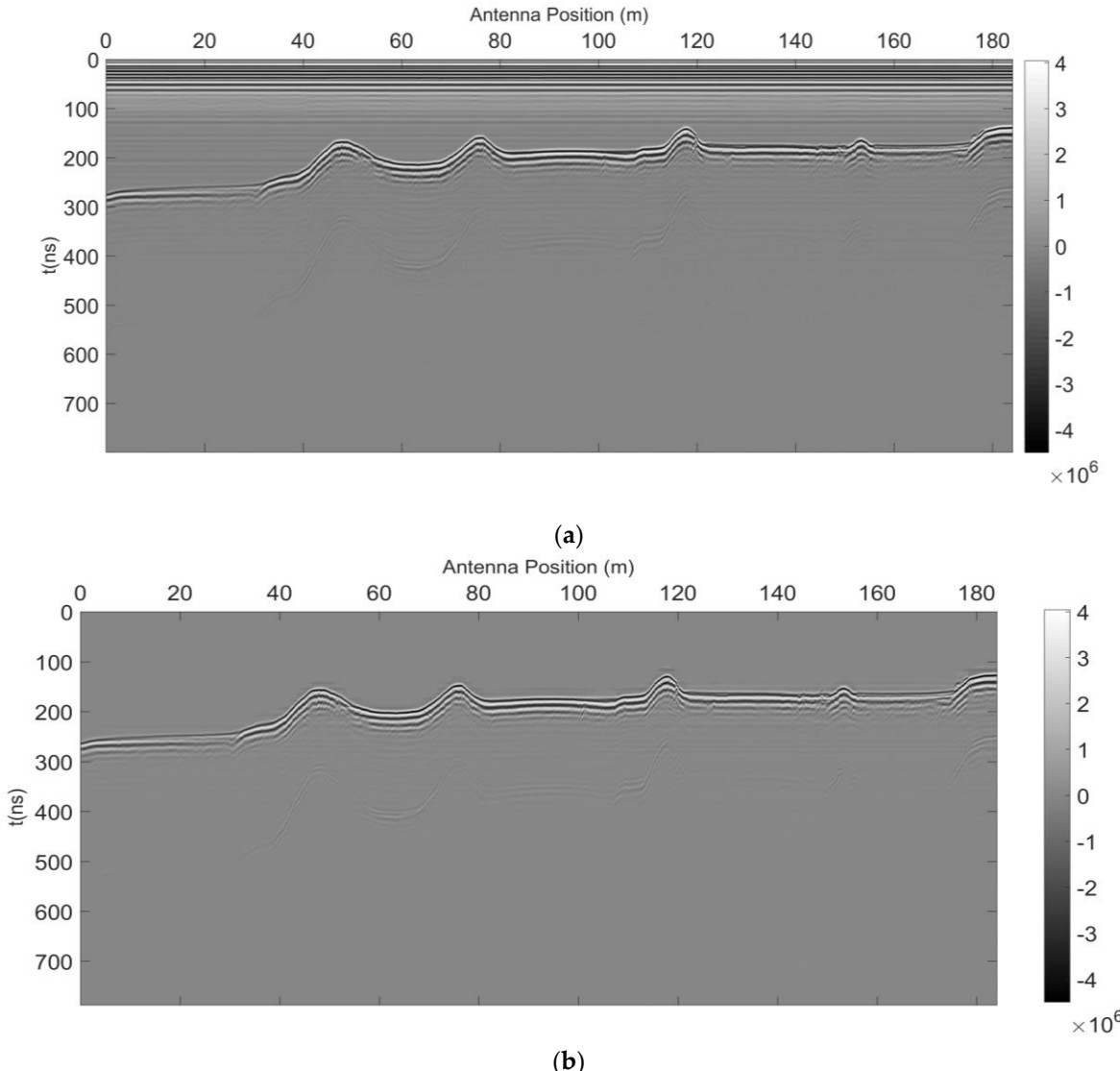

**Figure 6.** The measured B−scan profiles: (**a**) The raw B−scan profile; (**b**) The pre−processed B−scan profile.

### 3. Results

*3.1. Synthetic Datasets*

3.1.1. Underwater Horizontal Layered Model

It is crucial to select an appropriate global operator length and a white noise coefficient. Based on trials, these were set to 40 ns and 0.1%, respectively. Figure 7a shows the B-scan profile without the predicted multiples based on those in Figure 2c. Figure 7b illustrates the predicted surface-related multiples and Figure 7c shows the 10th A-scan data with and without the elimination of multiples. As seen in Figure 7a, primary waves from the water-layer subsurface and silt-layer subsurface were well developed and surface-related multiples were suppressed effectively. However, there were some residual multiples approximately 10 ns away from the primary wave in the silt-layer subsurface, which is equal to the travel time of this layer. Therefore, they are interval multiples. In Figure 7c, suppressed multiples were located mainly at 48 ns, 58 ns, and 68 ns. According to the suppressed multiples result, residual interval multiples were located at 44 ns.

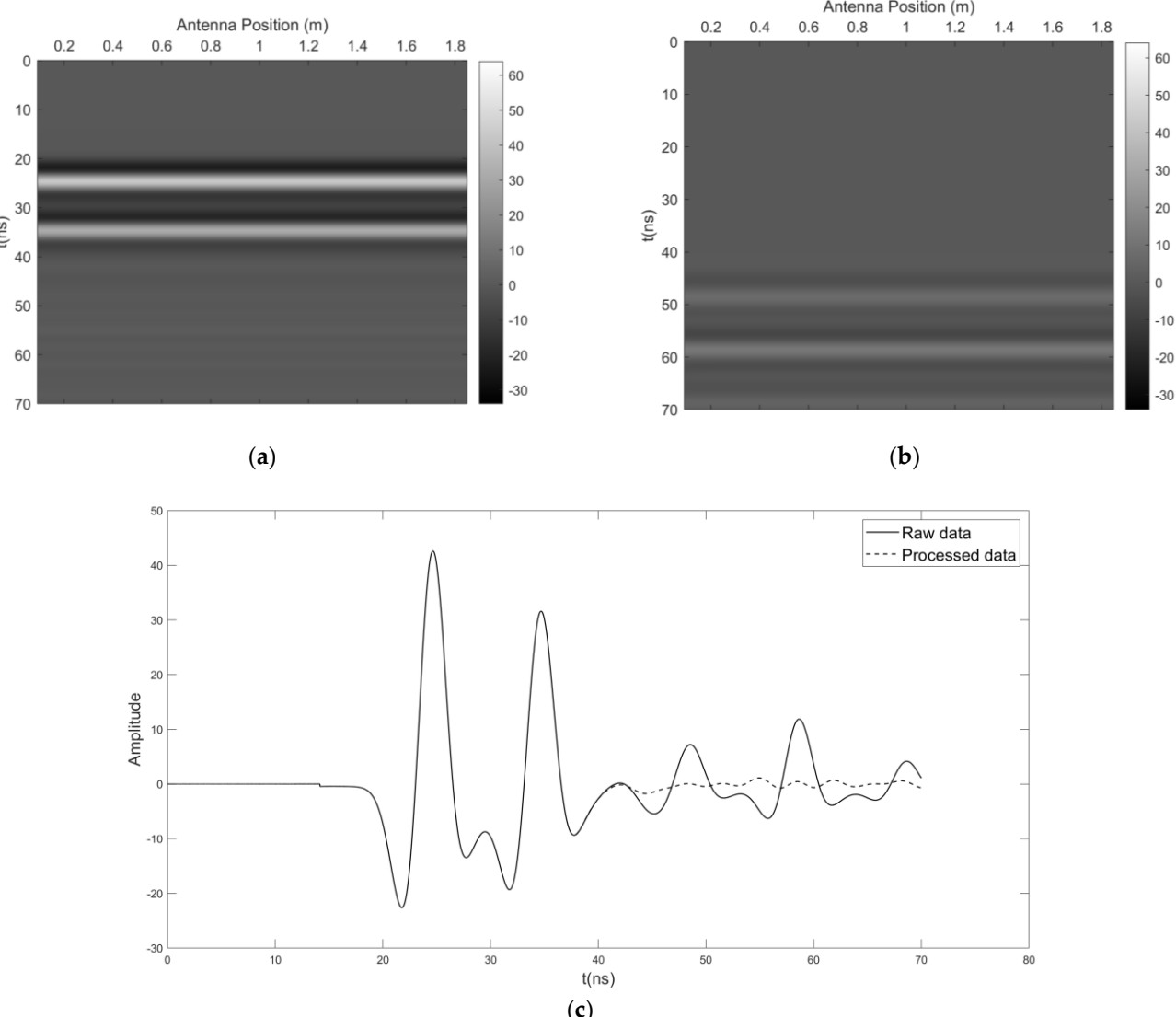

**Figure 7.** The SRME results: (**a**) The B−scan profile without the predicted multiples; (**b**) The B−scan profile containing predicted multiples; (**c**) A comparison of the 10th A−scan data with and without the elimination of multiples, in which the solid and dashed lines indicate the raw data and the processed data, respectively.

To validate the SRME method's effect on noise, the B-scan profile adds a Gaussian random noise level of 4%, as illustrated in Figure 2c. The length of the operator and the white noise coefficient were 40 ns and 0.1%. Figure 8a shows the multiples-eliminated B-scan profile from Figure 8c. Figure 8b presents the predicted surface-related multiples. Figure 8c compares the 10th A-scan signals. It indicates that Gaussian random noise had very little impact on the SRME results.

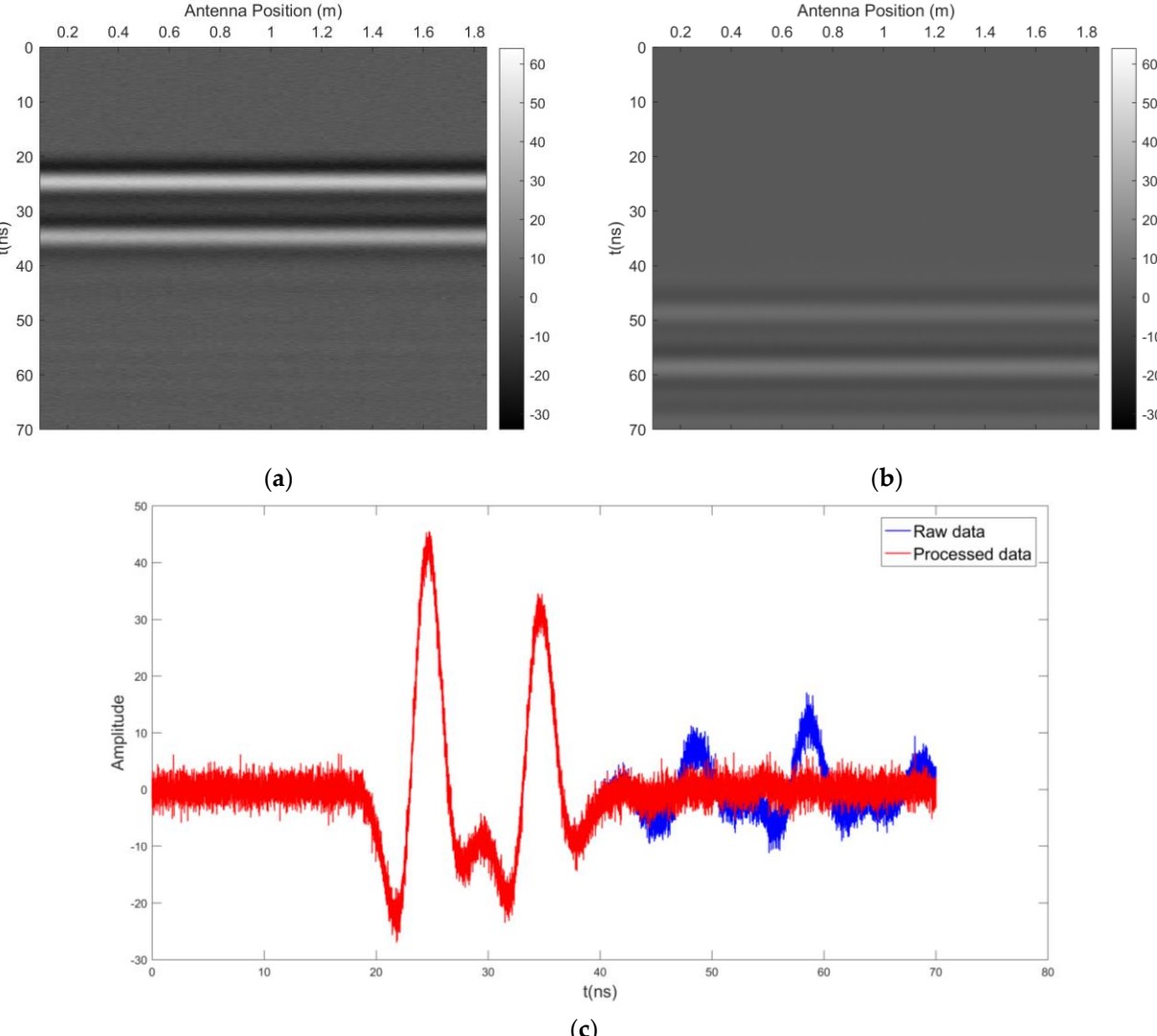

**Figure 8.** The SRME results with Gaussian random noise level of 4%: (**a**) The B−scan profile minus the predicted multiples; (**b**) The B−scan profile including the predicted multiples; (**c**) A comparison of the 10th A−scan data with and without the elimination of multiples, in which the red and blue lines indicate the raw and the processed data, respectively.

In order to further verify the effect of SRME, we used the predictive deconvolution method to suppress surface-related multiples. Figure 9 shows the results of this method under the same conditions. The prediction operator had a length of 35 ns, a white noise coefficient of 0.1%, and a prediction step of one reflection time at the underwater interface. Figure 10 shows the 10th A-scan data through the predictive deconvolution method.

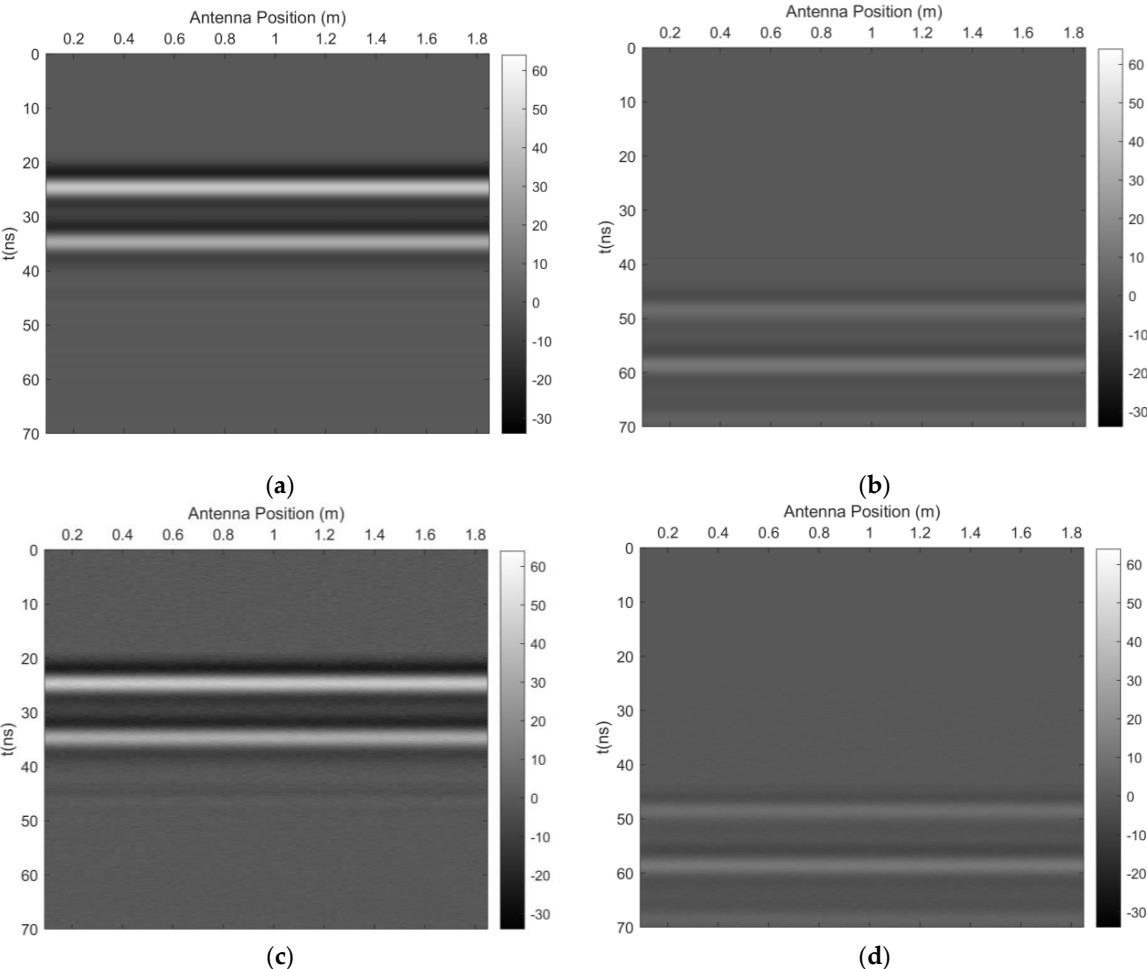

**Figure 9.** The predictive deconvolution method results: (**a**) The B−scan profile without the predicted multiples; (**b**) The B−scan profile containing the predicted multiples; (**c**) The B−scan profile without the predicted multiples with a Gaussian random noise level of 4%; (**d**) The B−scan profile including the predicted multiples with a Gaussian random noise level of 4%.

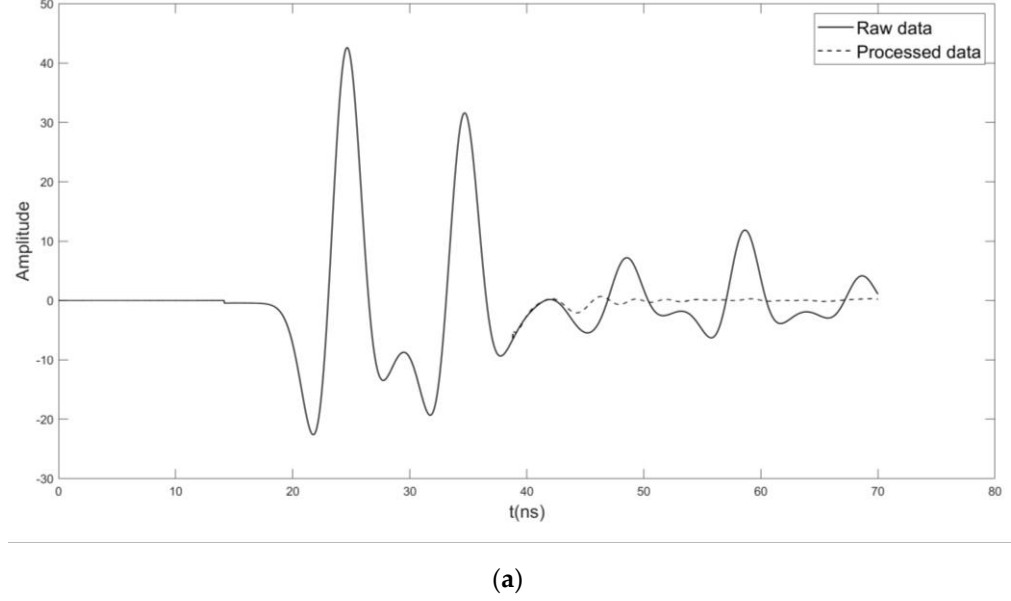

(**a**)

**Figure 10.** *Cont.*

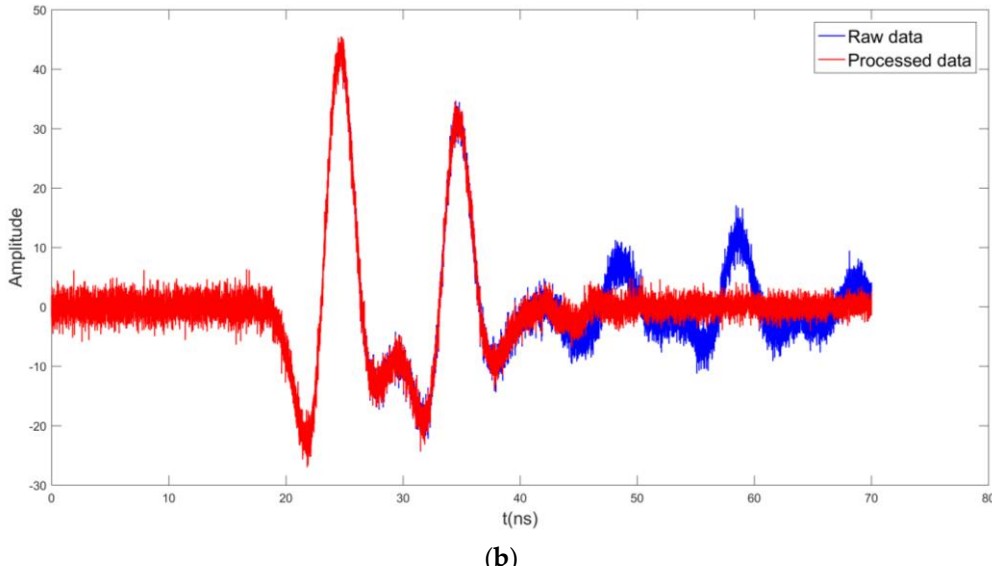

(**b**)

**Figure 10.** A comparison of the A−scan data: (**a**) without the Gaussian random noise level of 4%; (**b**) with the Gaussian random noise level of 4%, in which the red and blue lines indicate the raw and the processed data, respectively.

In order to quantitatively evaluate the SRME results, these single channel results and predictive deconvolution are represented by the signal to noise ratio (SNR), as shown in Table 3.

**Table 3.** Comparisons of SNR.

| Method | Noise | SNR (dB) |
|---|---|---|
| SRME | None | 11.1235 |
| Predictive deconvolution | None | 11.5178 |
| SRME | 4% Gaussian random noise | 11.1853 |
| Predictive deconvolution | 4% Gaussian random noise | 11.4822 |

3.1.2. Underwater Undulating Interface Model

The length of the global operator was set to 30 ns, while the white noise coefficient was identical to that of the horizontal layered model, 0.1%. Figure 11 shows all the SRME results including those determined under the Gaussian random noise level of 4% and without noise. As Figure 11a indicates, the interval multiple at 100 ns can be clearly delineated from the few residual multiples. In Figure 11b, the slope of the predicted multiples is not continuous. Figure 12 shows predictive deconvolution results similar to those in Figure 11, which had a predictive operator length of 80 ns and a white noise coefficient of 0.1%. In the same way, suppressions of multiple waves at 1 m and 4.5 m were evaluated by SNR, as seen in Table 4.

**Table 4.** Comparisons of SNR at 1 m and 4.5 m.

| Method | Location | Noise | SNR (dB) |
|---|---|---|---|
| SRME | 1 m | None | 12.6812 |
| | 4.5 m | None | 13.5961 |
| | 1 m | 4% Gaussian random noise | 12.4715 |
| | 4.5 m | 4% Gaussian random noise | 13.1093 |
| Predictive deconvolution | 1 m | None | 18.9891 |
| | 4.5 m | None | 13.857 |
| | 1 m | 4% Gaussian random noise | 17.5618 |
| | 4.5 m | 4% Gaussian random noise | 12.7906 |

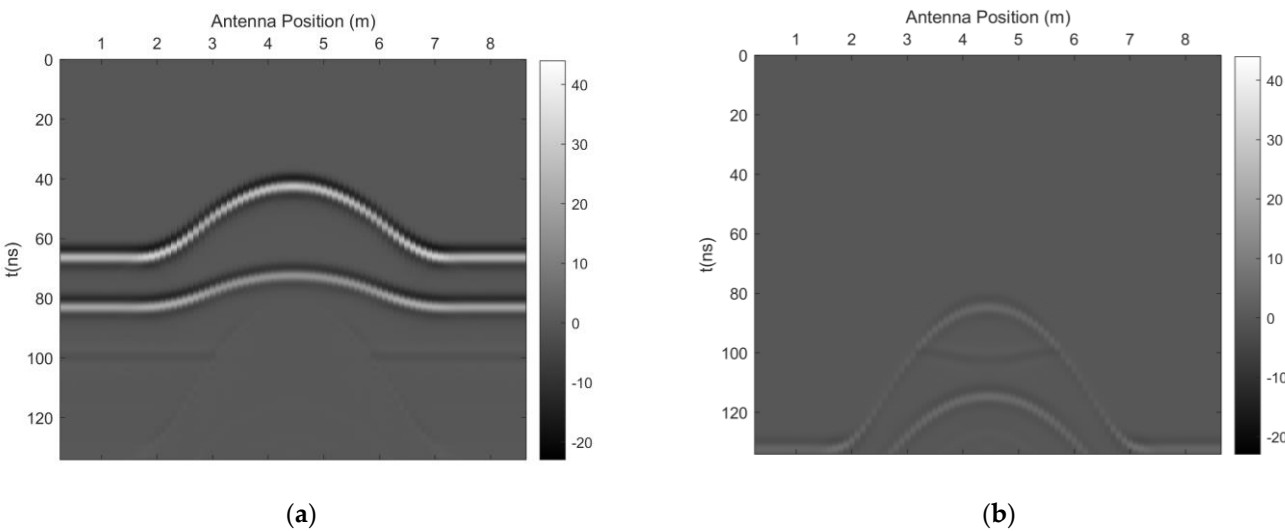

**Figure 11.** The SRME results: (**a**) The B−scan profile without the predicted multiples; (**b**) The B−scan profile including the predicted multiples; (**c**) The B−scan profile without the predicted multiples with a Gaussian random noise level of 4%; (**d**) The B−scan profile including the predicted multiples with a Gaussian random noise level of 4%.

**Figure 12.** *Cont.*

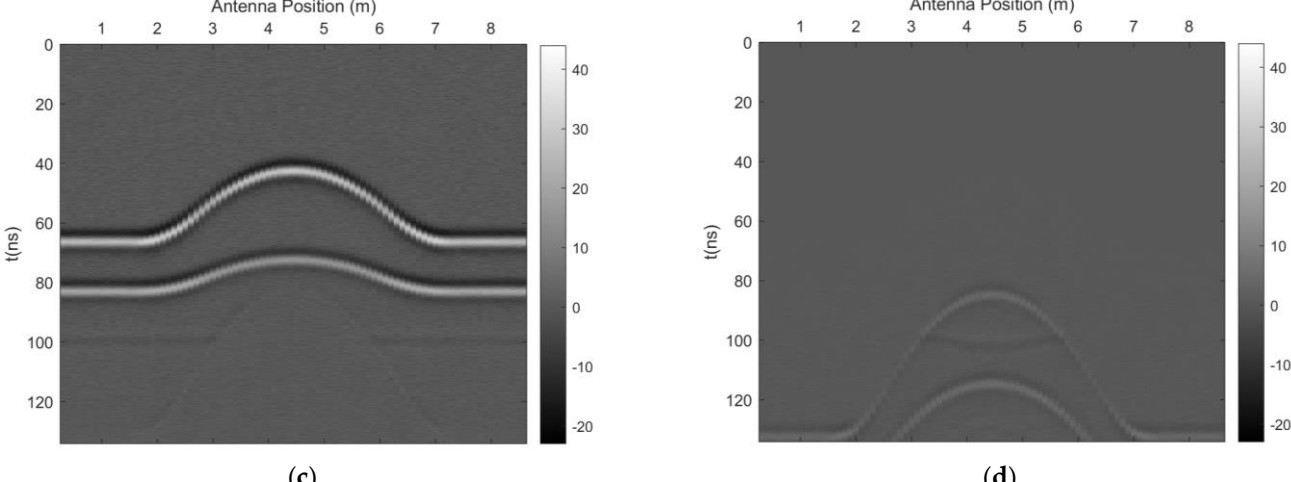

**Figure 12.** The predictive deconvolution results: (**a**) The B−scan profile without the predicted multiples; (**b**) The B−scan profile including the predicted multiples; (**c**) The B−scan profile without the predicted multiples with a Gaussian random noise level of 4%; (**d**) The B−scan profile including the predicted multiples with Gaussian random noise level of 4%.

### 3.2. Real Case Datasets

As shown in Figure 13a, the B-scan profile was processed to suppress multiples. Figure 13b displays the predicted multiples. The length of the global operator was set to 150 ns while the white noise coefficient was 0.1%. We found that the residual multiples, shown in Figure 13a, were mainly located in the undulating strata which correspond to the four increases. Moreover, some residual multiples ranged from 60 m to 70 m and came close to 400 ns in relatively flat terrain; thus, they should be interval multiples. We compared the pre-eliminated to the post-eliminated A-scan signals at 63 m, as seen in Figure 14a. We found multiples ranging from 390 ns to 430 ns and from 600 ns to 630 ns, respectively. The latter was slightly weaker. Similarly, we compared A-scan signals at 94 m, as shown in Figure 14b. The predicted multiples were found in the 40 ns from 340 ns to 380 ns, and the suppressing effect was significant; see Figure 15 for a more comprehensive comparison of all the predicted deconvolution results.

The SNR results of SRME and the predictive deconvolution method are presented in Table 5.

**Table 5.** Comparisons of SNR.

| Method | Location | SNR (dB) |
|---|---|---|
| SRME | 63 m | 0.8394 |
| Predictive deconvolution | 63 m | 0.3683 |
| SRME | 94 m | 1.2340 |
| Predictive deconvolution | 94 m | 0.2646 |

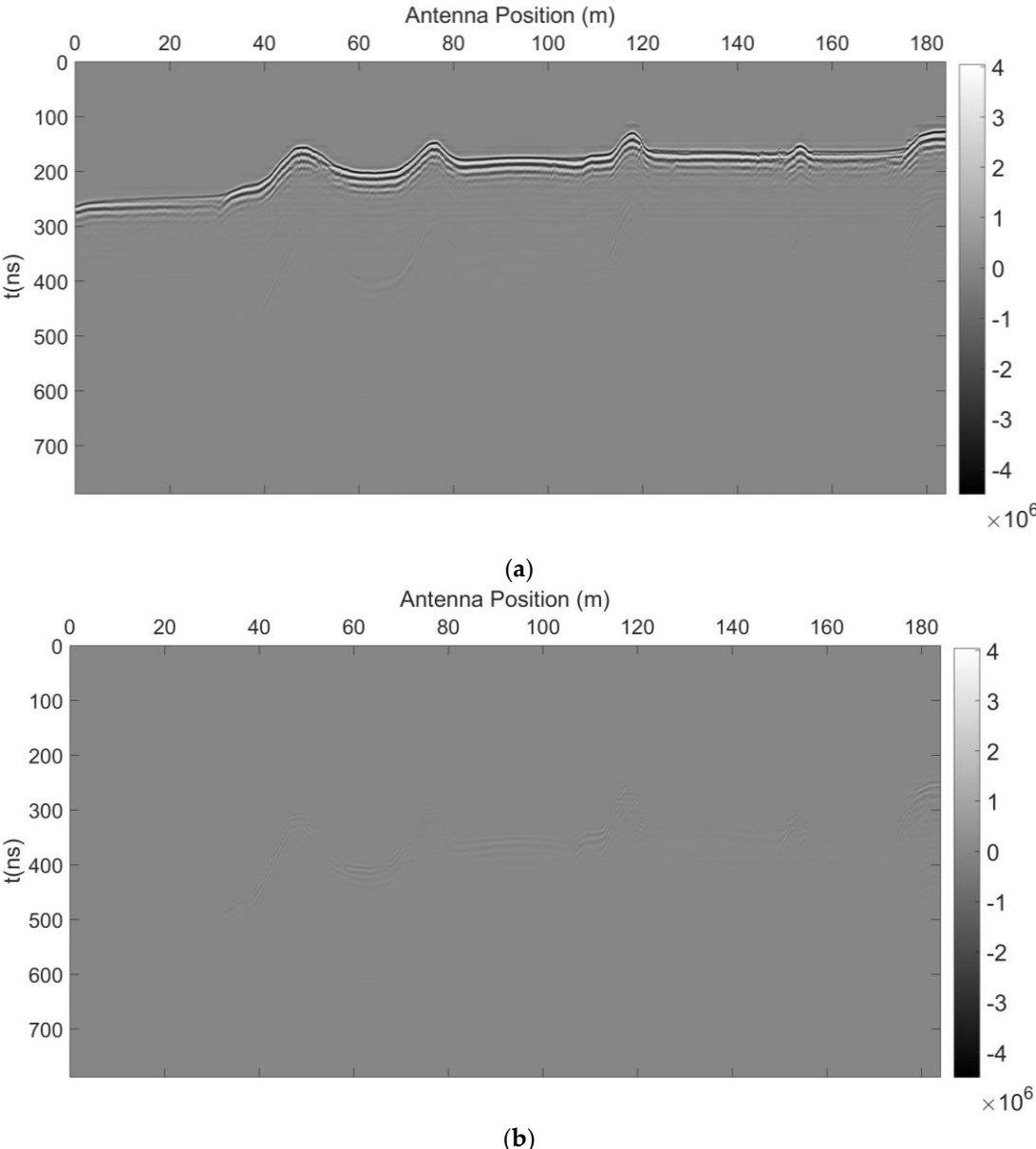

(a)

(b)

**Figure 13.** The SRME results: (**a**) The B−scan profile without the predicted multiples; (**b**) The B−scan profile including the predicted multiples.

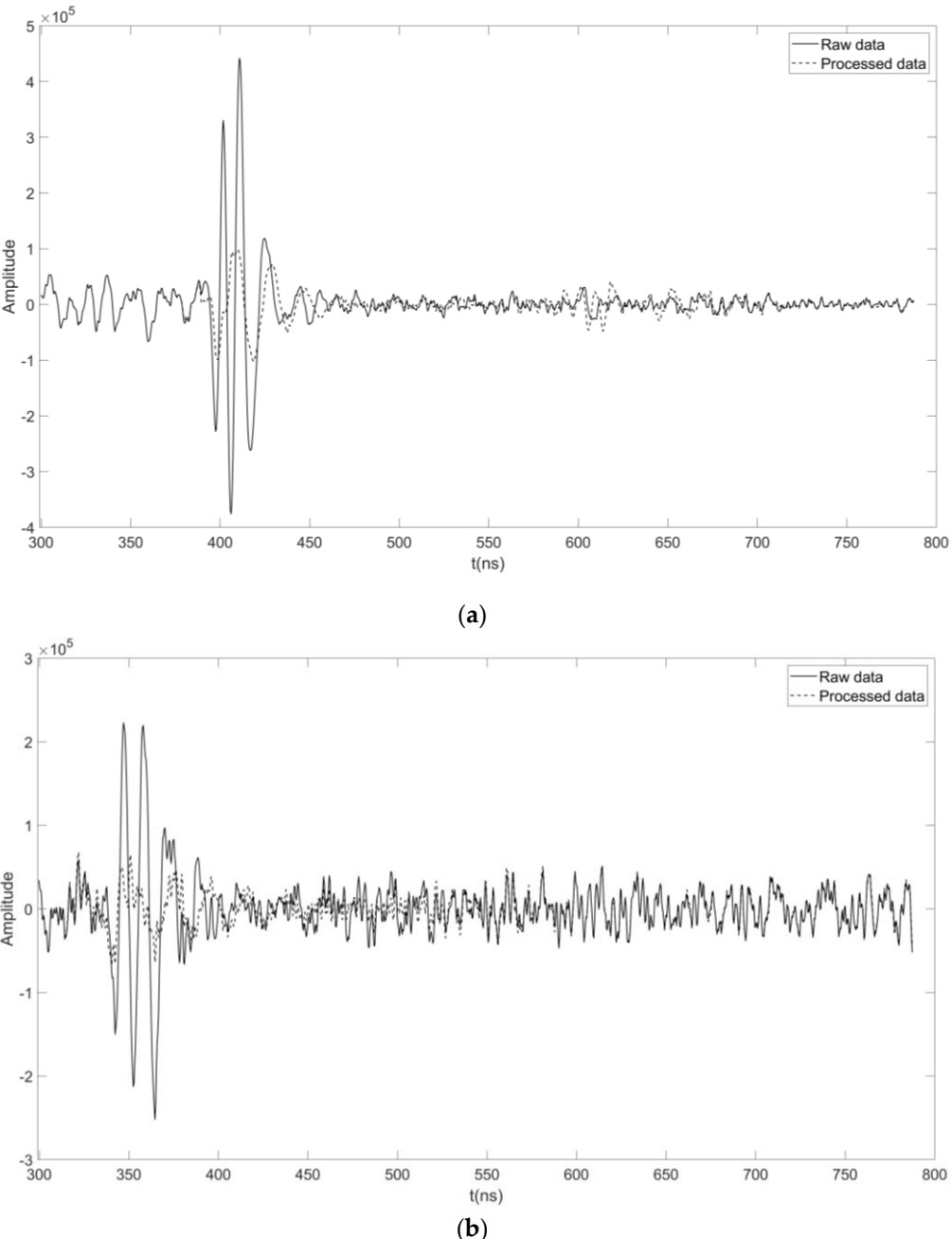

**Figure 14.** Comparisons of the A−scan data at: (**a**) 63 m; (**b**) 94 m. The solid and dashed lines represent the raw and the processed data, respectively.

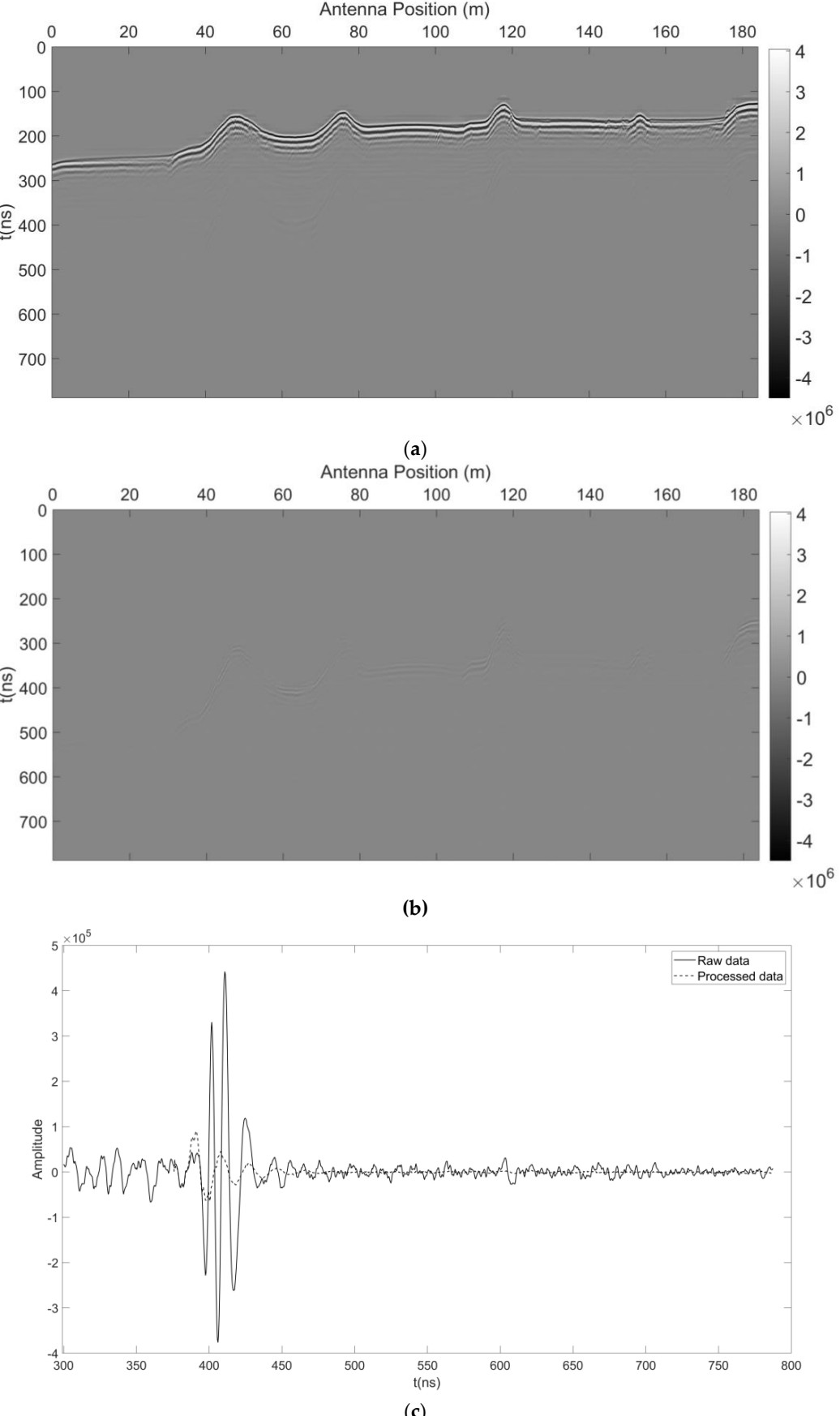

(**a**)

(**b**)

(**c**)

**Figure 15.** *Cont.*

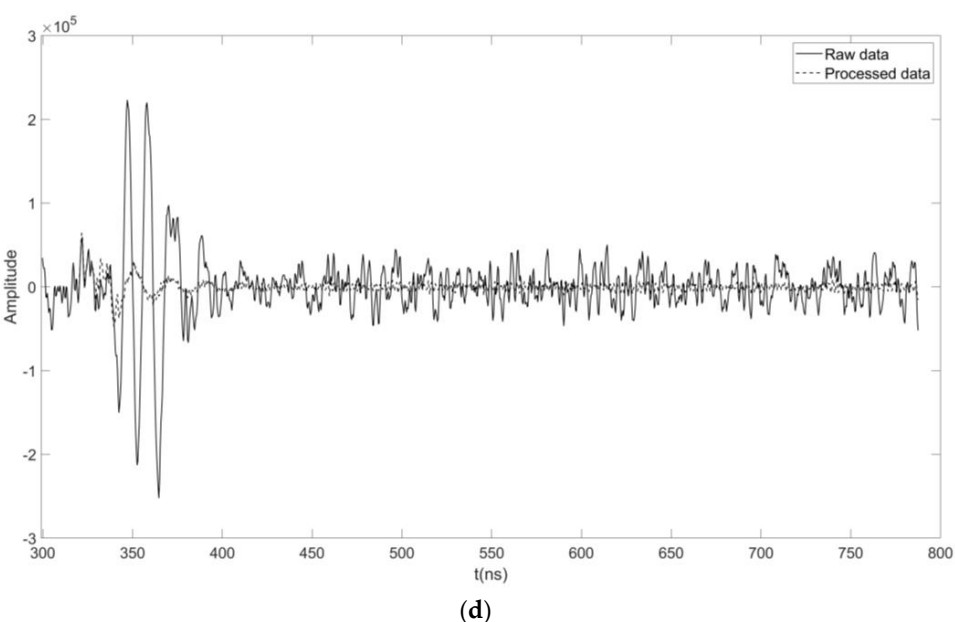

(**d**)

**Figure 15.** Predictive deconvolution results: (**a**) The B−scan profile without the predicted multiples; (**b**) Description of the contents of the second panel; (**c**) A comparison of the A−scan data at 63 m; (**d**) A comparison of the A−scan data at 94 m. The solid and dashed lines represent the raw and the processed data, respectively.

## 4. Discussion

The prediction deconvolution method and SRME have been utilized extensively in seismic research. However, the role of multiple suppression in GPR studies, especially in regard to water detection, has been largely ignored by scholars. Furthermore, compared to the predictive deconvolution method, SRME has received less attention in the field of water exploration. Tables 3 and 4 show that these techniques have similar suppression effects on multiple waves in water located in simple terrain; Table 5 further illustrates that for real world data, and SRME has a more suppressive effect on multiple waves generated by complex underwater terrain. This may be because real world underwater terrain is often more complex, and the predictive deconvolution method relies on prediction step size and prediction operators. This terrain makes it more difficult for the predictive deconvolution method to suppress multiple waves. For SRME, the relevant parameters include iteration number, global operator length, and white noise coefficient. The number of iterations and the white noise coefficient usually do not have a significant impact on the results of SRME. After analyzing the results of the experiments, we found that the optimal global operator length should be near the time of the first arrival of the underwater interface reflection. The Shanglin Lake site has simple underwater terrain, so GPR for sounding and mapping in this area has been successful. Due to the rapid attenuation of electromagnetic waves in underwater formations and deep water, sometimes multiple waves have very little impact on the processing and interpretation of actual data. In Figure 6, the water depth of flat stratum and the Penetration depth of electromagnetic wave are estimated to be 3.3 m and 2.7 m, respectively. The shallow underwater strata and anomalous bodies buried in the strata are not greatly affected by multiple waves. However, wave suppression is essential when researching real world complex and diverse underwater environments. Thus, it deserves more attention.

## 5. Conclusions

For this study, the SRME method was first utilized for GPR data processing because it can adaptively subtract and suppress predicted multiples by iteratively obtaining surface matching operators based on the principle of minimum energy. Compared to the predicted

deconvolution method, SRME is more effective at suppressing underwater free surface multiples from real world data. In addition, GPR can play an essential role in underwater sounding and underwater terrain mapping. Development of multiple waves is commonly found in underwater radar profiles. In areas with deep water depths, the impact of multiple waves on underwater strata and buried anomalous bodies is limited. However, to more extensively investigate complex underwater terrain and different water depths, research on multiple suppression is necessary. This paper provides a methodological basis for using GPR in underwater detection and for solving difficult problems in the field of water engineering.

**Author Contributions:** Conceptualization, R.S. and Y.Z.; methodology, R.S.; software, R.S.; validation, R.S., S.H., and H.C.; formal analysis, R.S. and H.C.; investigation, R.S. and S.C.; resources, Y.Z.; data curation, R.S.; writing—original draft preparation, R.S.; writing—review and editing, Y.Z. and S.H.; visualization, R.S. and H.C.; supervision, Y.Z. and S.G.; project administration, Y.Z., S.C. and S.G.; funding acquisition, Y.Z. All authors have read and agreed to the published version of the manuscript.

**Funding:** This research was sponsored by the Natural Science Foundation of China (No. 41774124).

**Institutional Review Board Statement:** Not applicable.

**Informed Consent Statement:** Not applicable.

**Data Availability Statement:** Data will be made available on request.

**Conflicts of Interest:** The authors declare no conflict of interest.

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
