# Peer review of "Surface-Related Multiples Elimination for Waterborne GPR Data"

_remotesensing, doi:10.3390/rs15133250_

Round 1

Reviewer 1 Report

In order to strengthen the present work, it is recommended that a broader frame of reference be employed. This could entail providing additional motivation for the study, as well as critically discussing the reasons for multiples in GPR sections. Additionally, the benefits of the proposed algorithm could be more thoroughly explored, potentially through a comparative analysis with other established methods, such as the singular value decomposition (SVD) or Karhunen-Lo´eve transform. Furthermore, a robust literature review could be included that addresses prior techniques utilized in the elimination of multiples in GPR sections, as well as the application of GPR technology in mapping the basements of lakes.

It is noted that the present manuscript primarily focuses on testing a particular method and presenting the corresponding results, as opposed to offering a thorough interpretation of the findings. To improve the manuscript, the authors are encouraged to offer a more comprehensive analysis and interpretation of their findings, potentially incorporating additional relevant literature to support their conclusions.

Most importantly, due to the high attenuation of GPR pulses within lake beds, the depth of penetration is limited, and the pulses rapidly weaken. Specifically, the depth of penetration for the 50 MHz antenna is less than one meter within the bed (https://doi.org/10.3997/2214-4609.20144857), which is similarly observed in the processed sections within the current manuscript. Consequently, processing for greater depths is deemed impractical, as such depths merely contain multiples with no practical use. How can the authors justify this issue?!

In the case of Figure5 (a) and (b), I see a dashed line on the top of first layer in GPR sections, could you please explain what the purpose of these lines are?!

How you chose the length of the global operator?! It looks like in your cases, it seems to be half of the time window (such as P8 L209), am I correct?! If yes, please explain and verify it in the manuscript.

Once again, reports and articles on the use of GPR to investigate subsurface water structures are very few and limited. However, the authors did not provide a complete overview of the previous works and gave very brief explanations about their case study and data collection. Presenting the details of the studied area, such as the percentage of water salinity or the type /size of bottom sediments, can make the article more interesting.

In the subchapter of ‘Principle of SRME Method’, where the main theory of the method used in this manuscript is explained, no references are mentioned. Are these formulas developed by the authors?!!

Presenting a flowchart in articles is remarkably interesting and welcome, but in this case, the steps of this flowchart have not been fully explained and only its steps have been mentioned again without more explanation. Since this section is the main part of the manuscript, it is suggested to extend this part completely and thoroughly (P3-4  L130-137).

P4 L132, what is your mean of filtering and denoising? Are they different? Could you please explain them and at lease mention the name of techniques you used in this step?!!

P4 L135-136, Please explain how these parameters effect the results.

P4 L152, ‘Gaussian random noise level of 4% …’ why you chose this level of random noise?!! Is this random noise level enough to evaluate the proposed algorithm?!! If yes, please mention a reference that verifies this level of noise for GPR data.

In the case of synthetic datasets, please use the same frequency of your case study, 100 MHz.

In the case of ‘2.3.1. Underwater horizontal layered mode’ section, why you first muted the direct waves and then added random noise?? it is better to mute the direct wave during processing steps. So, first, add random noise to your data and then process contaminated simulated data.

In the case of ‘2.3.2. Underwater undulating interface model’ section, it is better that authors follow the same steps as the first model and add random noise before applying processing techniques.

P7 L187, it is proposed to add a picture of the lake and some details about it. for example, the geology of the study area, or the importance and history of the lake.

In the case of the real case study, I wonder why the authors have not presented raw data for comparison, please show the raw data before applying any filter or correction.

P7 L193, in the case of ‘the suppression of noise’ it should be explained which kind of noise, and which filters had been used.

P7 L193-197, the sentence is ambiguous, please rewrite it.

P13 L279-280, ‘For underwater GPR data, there is only a local time window’. why?! is it your conclusion? if yes, please explain more.

P13-14 L281-284, Could you please explain and rewrite this sentence?!!

P14 L284, the sentence is left unfinished!!!!

P 14 L289-291, please use the quantitative measures that are a common way in scientific papers and reports such as SNR (Signal to Noise Ratio).

The discussion and conclusion sections are poorly written. The research results, significance and innovation point should be more accurate in it.

P13 L267, In Figure 11, it looks like you have attenuated the signal!!! Am I correct?!!

Figure 5 and10: The color bar is not well chosen. I suggest using the colour bar as in the synthetic model (gray) and making them equidistant.

In the case of A-scans (figures 6, 7, 9, and 11) you should ensure that all of your plots can be understood when printed in black & white, by e.g. changing line styles and plotting symbols, adding suitable labels etc., and avoid distinguishing features by colour alone in captions and legends.

Also, please ensure that all textual labels in figures are at least as large as the caption text; any smaller and they become too difficult to read.

 Good Luck

The manuscript should also undergo a proper revision with respect to language and wording. Issues are too numerous to list them all and are not included in this review. Also, this manuscript is presented more like a report than an article. 

An abbreviation is a shortened form of a word or phrase, to prevent its repetition in the text and to make the text easier to read. In this case, the first time a phrase or word is mentioned, its abbreviation must be mentioned, and after that, the abbreviation of that word is used. In this regard, please check the text thoroughly, especially regarding the ground penetrating radar (GPR).

P2 L77 principle,

P7 L186-199, most of sentence are simple, for example: The work site is shown as Figure 4. The pre-processed B-scan profile is shown in Figure 5-(a). As shown in Figure 5-(b), the length of the line is 184m. The dataset is composed of 4459 channels of data.

P8 L209, ‘Figure 6 shows’ should be deleted.

Author Response

Comment 1: In order to strengthen the present work, it is recommended that a broader frame of reference be employed. This could entail providing additional motivation for the study, as well as critically discussing the reasons for multiples in GPR sections. Additionally, the benefits of the proposed algorithm could be more thoroughly explored, potentially through a comparative analysis with other established methods, such as the singular value decomposition (SVD) or Karhunen-Lo´eve transform. Furthermore, a robust literature review could be included that addresses prior techniques utilized in the elimination of multiples in GPR sections, as well as the application of GPR technology in mapping the basements of lakes.

Response: Thank you very much for your comment. The application of GPR technology in mapping the basements of lakes has been added in the introduction. In fact, we have conducted some experiments on using filtering methods (including F-K filter and tau-p filter) to eliminate multiple waves, but the results are not ideal. The predictive deconvolution method has been mentioned in the introduction. But the literature about the predictive deconvolution method utilized in waterborne data is also few.

Comment 2: It is noted that the present manuscript primarily focuses on testing a particular method and presenting the corresponding results, as opposed to offering a thorough interpretation of the findings. To improve the manuscript, the authors are encouraged to offer a more comprehensive analysis and interpretation of their findings, potentially incorporating additional relevant literature to support their conclusions. 

Response: Your suggestion really means a lot to us. For a more comprehensive and interpretation, we add the other multiples suppression method, i.e. the predictive deconvolution method. Also, we adopt the quantitative method of SNR to compare the multiples-suppression effect of SRME with that of the predictive deconvolution method.

Comment 3: Most importantly, due to the high attenuation of GPR pulses within lake beds, the depth of penetration is limited, and the pulses rapidly weaken. Specifically, the depth of penetration for the 50 MHz antenna is less than one meter within the bed (https://doi.org/10.3997/2214-4609.20144857), which is similarly observed in the processed sections within the current manuscript. Consequently, processing for greater depths is deemed impractical, as such depths merely contain multiples with no practical use. How can the authors justify this issue?!

Response: Thank you very much for your comment. According to some references, for example, https://doi.org/10.1023/A:1007920816271, authors referred to lake bottom multiples and thought that there must be a good sub-bottom penetration or the shallow water depth. In general, the propagation velocity of electromagnetic wave in fresh water is 0.033m/ns. If the sediments are saturated silt or saturated clay (both of propagation velocities are considered as 0.09m/ns), the ratio of depth of penetration to water depth would have to be less than 3:1 before interference between the primary and the multiple would occur. However, the propagation velocity of lake ice can reach 0.16m/ns. It means that the ratio of depth of penetration to lake ice thickness would be less than 1. So we think that this research about underwater multiples suppression is significant.

Comment 4: In the case of Figure5 (a) and (b), I see a dashed line on the top of first layer in GPR sections, could you please explain what the purpose of these lines are?!

Response: Thanks for your suggestions. Sorry to make such a mistake. The mistake has been corrected.

Comment 5: How you chose the length of the global operator?! It looks like in your cases, it seems to be half of the time window (such as P8 L209), am I correct?! If yes, please explain and verify it in the manuscript.

Response: Thank you very much for your comment. The length of the global operator is not half of the time window. Actually, we experimented a lot and found that SRME with the operator’s length of 5ns rarely suppressed multiples, and the effect of multiples suppression reached the peak with the operator’s length of 40ns. The longer operator also achieved the same effect. However, the longer length means the more time spent. For the computation efficiency, we set the lengths of the global operator of the two models and real case to 40ns, 30ns and 150ns.

Comment 6: Once again, reports and articles on the use of GPR to investigate subsurface water structures are very few and limited. However, the authors did not provide a complete overview of the previous works and gave very brief explanations about their case study and data collection. Presenting the details of the studied area, such as the percentage of water salinity or the type /size of bottom sediments, can make the article more interesting.

Response: According to your suggestion, the more details of the studied area have been incorporated into the introduction of the studied area.

Comment 7: In the subchapter of ‘Principle of SRME Method’, where the main theory of the method used in this manuscript is explained, no references are mentioned. Are these formulas developed by the authors?!!

Response: According to your suggestion, Relative references have been mentioned in the subchapter of ‘Principle of the SRME Method’.

Comment 8: Presenting a flowchart in articles is remarkably interesting and welcome, but in this case, the steps of this flowchart have not been fully explained and only its steps have been mentioned again without more explanation. Since this section is the main part of the manuscript, it is suggested to extend this part completely and thoroughly (P3-4  L130-137).

Response: Thank you very much for your comment. More details have been complemented in the subchapter of ‘workflow’.

Comment 9: P4 L132, what is your mean of filtering and denoising? Are they different? Could you please explain them and at lease mention the name of techniques you used in this step?!!

Response: Thank you very much for your comment. The used filtering method is FIR (Finite Impulse Response) filter. The denoising method is different from the filtering method. The motivation of the denoising method is to eliminate the interference of noise from water-bodies for the effect of SRME. For achieving the denoising method, the waterbodies data need to be changed into some weak Gaussian random noise instead of 0.

Comment 10: P4 L135-136, Please explain how these parameters effect the results.

Response: Thanks for your comments. The critical parameters include the global operator length, the iteration time and the white noise coefficient, i.e. the stability coefficient. For out study, the best global operator length was not fixed and was no more than half the time window. There were generally no more than five iterations. The value of the white noise coefficient varies from 0.1% to 1%, which had very little impact on the SRME results.

Comment 11: P4 L152, ‘Gaussian random noise level of 4% …’ why you chose this level of random noise?!! Is this random noise level enough to evaluate the proposed algorithm?!! If yes, please mention a reference that verifies this level of noise for GPR data.

Response: Thank you very much for your comment. The pre-processing step of numerical experiments includes time-zero correction, direct-wave muting and removal of background. The added Gaussian random noise level is 4% of the maximum amplitude. Moreover, the real case data need to be filtered to suppress the random noise. The residual noise level of the processed data is not high. So we choose the Gaussian random noise level of 4%.

Comment 12: In the case of synthetic datasets, please use the same frequency of your case study, 100 MHz.

Response: Thanks for your suggestion. The frequency of the synthetic datasets has been changed into 100MHz.

Comment 13: In the case of ‘2.3.1. Underwater horizontal layered mode’ section, why you first muted the direct waves and then added random noise?? it is better to mute the direct wave during processing steps. So, first, add random noise to your data and then process contaminated simulated data.

Response: Thank you very much for your comment. The synthetic data have been re-processed as your suggestion.

Comment 14: P7 L187, it is proposed to add a picture of the lake and some details about it. for example, the geology of the study area, or the importance and history of the lake.

Response: According to your suggestion, the panorama of the Shanglin Lake has been added. Moreover, the history of the Shanglin Lake and its significance are introduced thoroughly.

Comment 15: In the case of the real case study, I wonder why the authors have not presented raw data for comparison, please show the raw data before applying any filter or correction.

Response: Thanks for your correction. So sorry to make such a mistake. The raw data has been added for comparison.

Comment 16: P7 L193, in the case of ‘the suppression of noise’ it should be explained which kind of noise, and which filters had been used.

Response: Thank you very much for your comment. The filtering methods are used to suppress the ambient noise, which usually include the removal of the global background and the finite impulse response (FIR) filter. In this paper, the denoising method is utilized for eliminating the in-fluence of residual noise in water body.

Comment 17: P7 L193-197, the sentence is ambiguous, please rewrite it.

Response: Thanks for your correction. Sorry to make such a mistake. The sentence has been rewritten.

Comment 18: P13 L279-280, ‘For underwater GPR data, there is only a local time window’. why?! is it your conclusion? if yes, please explain more.

Response: Thanks for your correction. Sorry to make such a mistake. The discussion and conclusion have been rewritten.

Comment 19: P13-14 L281-284, Could you please explain and rewrite this sentence?!!

Response: Thank you very much for your comment. This sentence has been rewritten. In this paper, we used the global operator for SRME. However, the processed profiles remain some residual multiples. The residual multiples are mainly internal multiples. So Verschuur and Berkhout (DOI: 10.1190/1.1444262.) pointed out that the SRME method can also perform local elimination within the local window after global elimination.

Comment 20: P14 L284, the sentence is left unfinished!!!!

Response: Thanks for your correction. Sorry to make such a mistake. All the mistakes have been corrected.

Comment 21: P 14 L289-291, please use the quantitative measures that are a common way in scientific papers and reports such as SNR (Signal to Noise Ratio).

Response: Thanks for your comments. According to your suggestion, we added the quantitative measures with SNR.

Comment 22: The discussion and conclusion sections are poorly written. The research results, significance and innovation point should be more accurate in it.

Response: Thanks for your comments. The discussion and conclusion sections have been rewritten.

Comment 23: P13 L267, In Figure 11, it looks like you have attenuated the signal!!! Am I correct?!!

Response: Thanks for your comments. In fact, the signal is not attenuated. This figure is about the comparison between the pre-processed data and multiples-suppressed data. The pre-processed data was attenuated due to SRME.

Comment 24: Figure 5 and10: The color bar is not well chosen. I suggest using the colour bar as in the synthetic model (gray) and making them equidistant.

Response: According to your suggestion, the colour bars of the two figures have been changed into gray.

Comment 25: In the case of A-scans (figures 6, 7, 9, and 11) you should ensure that all of your plots can be understood when printed in black & white, by e.g. changing line styles and plotting symbols, adding suitable labels etc., and avoid distinguishing features by colour alone in captions and legends.

Response: Thanks for your comments. In Figure 8(c) and Figure 10(b), a Gaussian random noise level of 4% is added. If both figures are plotted in black & white, the difference is not clear.

Comment 26: Also, please ensure that all textual labels in figures are at least as large as the caption text; any smaller and they become too difficult to read.

Response: Thanks for your correction. Sorry to make such a mistake. All textual labels in figures have been corrected.

Comment 27: The manuscript should also undergo a proper revision with respect to language and wording. Issues are too numerous to list them all and are not included in this review. Also, this manuscript is presented more like a report than an article.

Response: According to your suggestion, the manuscript has been thoroughly revised and re-polished by a native English speaker.

Comment 28: An abbreviation is a shortened form of a word or phrase, to prevent its repetition in the text and to make the text easier to read. In this case, the first time a phrase or word is mentioned, its abbreviation must be mentioned, and after that, the abbreviation of that word is used. In this regard, please check the text thoroughly, especially regarding the ground penetrating radar (GPR).

Response: Thanks for your correction. Sorry to make such a mistake. All the abbreviation has been corrected especially the ground penetrating radar.

Comment 29: P2 L77 principle,

Response: Thanks for your correction. Sorry to make such a mistake. The mistake has been corrected.

Comment 30: P7 L186-199, most of sentence are simple, for example: The work site is shown as Figure 4.The pre-processed B-scan profile is shown in Figure 5-(a). As shown in Figure 5-(b), the length of the line is 184m. The dataset is composed of 4459 channels of data.

Response: Thanks for your comments. Sentences have been modified.

Comment 31: P8 L209, ‘Figure 6 shows’ should be deleted.

Response: Thanks for your correction. Sorry to make such a mistake. The words have been deleted.

Reviewer 2 Report

Dear Authors,

Congratulations on the paper. The GPR community was needing this procedure to identify deeper targets under multiples.

Please find my comments attached.

Very minors comments.

Author Response

Comment 1: In general I believe the article is well written in well explained. It has a great application to the real field. In several cases, it is complicated to identify anomalies under multiples of layers of high contrast.

I would recommend to include a case where would have a target (as shown in red) under this layer and at the same position of the multiples to evaluate the effectiveness of the process. Maybe in a second version of this paper. I don’t consider it mandatory.

Response: Thanks for your kind comments. We believe that even if the surface-related multiples and the target signals overlap, the predicted surface-related multiples will not differ greatly from the actual multiples, and even multiple suppression will not affect the target signal recognition. So we still choose the original horizontal model.

Comment 2: I would recommend the authors to avoid personal terms such as WE DID. Instead could be it was done.

Response: Thanks for your comments. This paper has been modified. We believe your suggestion is very important.

Comment 3: Page 72 surface-related multiples of SRME while a real case dataset was processed. In order I believe there is a mistake in this sentence. Please review.

Response: Thanks for your correction. This sentence has been corrected according to your suggestions.

Comment 4: Page 284 ture, an incomplete reflection of water surface, and the mi the sentence is not finished.

Response: Thanks for your correction. Sorry to make such a mistake. The mistake has been corrected.

Comment 5: The authors used both antenna and antena (in some figures). I would select only term.

Response: Thanks for your comments. This paper has been corrected according to your suggestions. The antenna is used in those figures.

Comment 6: The text of the figure (subtitle) sometimes finished with full stop and sometimes there is nothing.

Response: Thanks for your correction. This paper has been corrected according to your suggestions.

Comment 7: I would add AMPLITUDE on the right of the figure.

Response: Thanks for your comments. Sometimes we are used to add it on the left of the figure. We think your suggestion is good.

Comment 8: The paper quality is not represented in the conclusions. I would develop and explain that the process allow identify targets that might be masked by the presence of multiples.

Response: Thanks for your comments. In the discussion and conclusion, we explain that the impact of multiple waves on underwater strata and buried anomalous bodies is limited in areas with deep water depths. The shallow underwater strata and anomalous bodies buried in the strata are not greatly affected by multiple waves. But wave suppression is essential when researching real world complex and diverse underwater environments.

Round 2

Reviewer 1 Report

The manuscript titled "Surface-Related Multiples Elimination for Waterborne GPR Data" that was submitted to the Journal of Remote Sensing has undergone a comprehensive review once more. It gives me pleasure to suggest that it be accepted for publication.

I commend the authors on their diligent work and commend them for their valuable contribution to the field.

Sincerely,

Dr. Saeed Parnow

Reviewer